# Neuropsychiatric Manifestations of Mast Cell Activation Syndrome and Response to Mast-Cell-Directed Treatment: A Case Series

**DOI:** 10.3390/jpm13111562

**Published:** 2023-10-31

**Authors:** Leonard B. Weinstock, Renee M. Nelson, Svetlana Blitshteyn

**Affiliations:** 1Independent Researcher, Specialists in Gastroenterology, St. Louis, MO 63141, USA; 2Department of Neurology, Jacobs School of Medicine and Biomedical Sciences, University at Buffalo, Buffalo, NY 14203, USA; reneenel@buffalo.edu (R.M.N.); sb25@buffalo.edu (S.B.); 3Dysautonomia Clinic, Williamsville, NY 14221, USA

**Keywords:** anxiety, depression, dysautonomia, mast cell activation syndrome, panic disorder, POTS

## Abstract

Mast cell activation syndrome (MCAS) is an immune disease with an estimated prevalence of 17%. Mast cell chemical mediators lead to heterogeneous multisystemic inflammatory and allergic manifestations. This syndrome is associated with various neurologic and psychiatric disorders, including headache, dysautonomia, depression, generalized anxiety disorder, and many others. Although MCAS is common, it is rarely recognized, and thus, patients can suffer for decades. The syndrome is caused by aberrant mast cell reactivity due to the mutation of the controller gene. A case series is presented herein including eight patients with significant neuropsychiatric disorders that were often refractory to standard medical therapeutics. Five patients had depression, five had generalized anxiety disorder, and four had panic disorder. Other psychiatric disorders included attention-deficit hyperactivity disorder, obsessive compulsive disorder, phobias, and bipolar disorder. All eight patients were subsequently diagnosed with mast cell activation syndrome; six had comorbid autonomic disorders, the most common being postural orthostatic tachycardia syndrome; and four had hypermobile Ehlers-Danlos syndrome. All patients experienced significant improvements regarding neuropsychiatric and multisystemic symptoms after mast-cell-directed therapy. In neuropsychiatric patients who have systemic symptoms and syndromes, it is important to consider the presence of an underlying or comorbid MCAS.

## 1. Introduction

Mast cell activation syndrome (MCAS) presents with heterogenous multisystemic inflammatory and allergic manifestations [1,2,3]. MCAS is characterized by patterns of aberrant mast cell (MC) overactivity [2]. Mast cell activation disease (MCAD), which includes MCAS and mastocytosis, is associated with neuropsychiatric disorders, including various types of dysautonomia, neuropathy (including small fiber neuropathy), myalgia, migraine, headache, cognitive dysfunction, restless legs syndrome, sleep disturbance, non-pulsatile tinnitus, depression, generalized anxiety, and panic attacks [2,4]. MCAS is the most common variant of MCAD and has an estimated prevalence of 17% in the general population [5]. Despite a significant prevalence, this hyperactive immune disorder is usually not considered in the differential diagnosis in patients with multisystemic symptoms [1,6]. This is in part due to its relatively recent discovery (2007) and it is generally not included in medical school curriculum [7].

The heterogeneity of MCAS is vast, with symptoms and syndromes across various domains including constitutional, dermatologic, ophthalmologic, otologic, oropharyngeal, lymphatic, pulmonary, cardiovascular, gastrointestinal, genitourinary, musculoskeletal, neurologic, psychiatric, metabolic, hematologic, and immunologic systems (Table 1) [2]. Patients with unrecognized, undiagnosed MCAS will often see multiple specialists and ultimately stop reporting symptoms owing to poor experiences with the medical system. Unfortunately, these patients are at risk of being misdiagnosed with somatization disorder or Munchausen’s syndrome.

MCs can be located adjacent to blood vessels along the blood–brain barrier (BBB) and interact with microglia, astrocytes, and blood vessels through stored or synthesized neuroactive mediators [8]. As the effector cells of the innate immune system, MCs are first to respond to injury, releasing proinflammatory signals to which microglia respond through the production of cytokines, chemokines, glutamate, and reactive oxygen species [9]. Mature MCs can migrate to the brain from the vascular system, and evidence suggests that MCs themselves may disrupt the integrity of the BBB [8]. Signaling between MCs and microglia modulates immunologic responses to inflammation, infection, trauma, and stress. In the setting of prolonged neuroinflammation, these controls may be less effective, and aberrant inflammatory responses may ensue.

The severity of symptoms ranges from mild to life-threatening when anaphylaxis is present. The degree of morbidity is related to the quantity of the affected mast cells (MCs), the number of mutations within the mast cell (MC) lineage, the specific organ involvement, the presence of comorbid postural orthostatic tachycardia syndrome (POTS) and hypermobile Ehlers-Danlos syndrome (hEDS), and the impact of triggers [10,11,12,13]. There are many MC triggers, including diet, stress, estrogens, excipients, and a variety of infections [14,15]. The long-lasting, often hidden triggers on which treatment can have a significant impact include small intestinal bacterial overgrowth, mycotoxin and chemical exposures, and heavy metal toxicity [16,17,18,19]. We present a case series of patients diagnosed with refractory neuropsychiatric disorders who were subsequently diagnosed with MCAS and whose conditions improved with MC-directed treatment.

## 2. Materials and Methods

This is a retrospective case series of eight patients who experienced chronic neuropsychiatric symptoms and disorders and were subsequently diagnosed by the authors with MCAS. We observed that these patients experienced significant improvement regarding their psychiatric symptoms when their MCAS was treated with MC-directed therapy. Criteria for inclusion in this case series were a diagnosis of a psychiatric disorder by a primary care physician and/or a psychiatrist and a diagnosis of MCAS (Table 2) [6]. Exclusion criteria included mastocytosis or having symptoms and signs best explained by a medical condition other than MCAS. Comorbid syndromes including autonomic disorders, such as postural orthostatic tachycardia syndrome (POTS), and joint hypermobility syndromes, such as hypermobile Ehlers-Danlos syndrome (hEDS), were assessed. All subjects gave their informed consent for inclusion before they participated in the study. The study was conducted in accordance with the Declaration of Helsinki. The study was approved by the University at Buffalo Jacobs School of Medicine Institutional Review Board (STUDY00006936).

Although many FDA-approved medicines have been studied and used for MCAS, there is no FDA-approved protocol for MCAS [14]. As part of our standard approach, we advise a 3-week trial on a gluten-free, dairy-free, and low-histamine diet. Medical therapy starts with a combination of non-sedating histamine receptor 1 and histamine receptor 2 blockers twice daily [9]. If the response is inadequate or the patients have significant symptoms, they receive additional over-the-counter MC stabilizing agents such as vitamins C and D and quercetin, a flavonoid. For those with extensive symptoms, they are administered the Step 1 MC-directed therapy, which includes antihistamines, vitamins C and D, quercetin, and the addition of low-dose naltrexone (LDN). The use of LDN has been reported to be effective in treating MCAS and depression [20,21]. The administration of additional medicines including immune modulators and chemotherapy are used for refractory MCAS [14,22].

## 3. Results

The subjects included seven females and one male with a mean age of 36 years (ages ranged from 18 to 71 years). Prior neuropsychiatric diagnoses, medical therapy, past medical history, new diagnoses, and the outcomes of MC-directed therapy are outlined in Table 3. Five had depression and one had bipolar disease. Two of these patients had attempted suicide as teenagers. Five had generalized anxiety disorder and four had panic disorder. Seven subjects had additional disorders: attention-deficit/hyperactivity disorder, obsessive–compulsive disorders, phobias, Tourette’s syndrome, and narcolepsy.

We subsequently diagnosed significant disorders: MCAS in eight subjects, POTS in four, labile blood pressure in two, inappropriate sinus tachycardia in one, neurocardiogenic syncope in one, migraine in two, and restless legs syndrome in two. Four subjects had hEDS. All subjects’ symptoms improved with MC-directed therapy, as shown in Table 3. Six subjects experienced significant improvement regarding their psychiatric disorder so they could return to work, high school, or college. One woman could now function well as a homemaker. Her daily suicidal ideation, which had persisted for decades, also ceased. One subject improved enough to work part time.

### 3.1. Illustrative Case: Patient 1

The patient was a 47-year-old physician who was healthy except for episodes of anaphylaxis in response to nuts in her childhood and adolescence. In her twenties, she experienced symptoms of weight loss, lymphadenopathy, night sweats, and fatigue, and at one point, she was thought to have lymphoma, which was subsequently ruled out. She was treated with amoxicillin for presumed bacterial infection, which caused a rash. At that time, her treating physicians thought she may have had mononucleosis. Subsequently, she experienced two bouts of shingles. She experienced no psychiatric symptoms and received no psychiatric diagnosis in childhood, adolescence, or in her twenties and felt that she was healthy and athletic prior to her first pregnancy. During her first pregnancy at age 35, she developed severe nausea associated with weight loss and was diagnosed with hyperemesis gravidarum. During her first pregnancy, she was treated with IV hydration via a central venous line on a temporary basis with good results, and the central venous line was removed toward the end of her pregnancy. She subsequently had uneventful labor, vaginal delivery, and postpartum period and was able to return to work full time without symptoms. She developed a severe phobia of blood 1.5 years post-partum, along with generalized anxiety and obsessive symptoms. Her second pregnancy was at 38 years of age, which was again associated with nausea and food intolerances, but she was able to maintain her weight and did not require a central venous line. However, she did receive intravenous hydration every few weeks through a peripheral line for symptoms thought to be due to dehydration. After an uneventful delivery, her phobia of blood worsened, and she developed compulsive rituals as well as fatigue. A few years later, she developed recurrent rashes, periorbital swelling, flushing episodes (Figure 1), rapid heart rate upon standing or with minimal exertion, and chronic constipation. She also developed presyncope associated with diarrhea and tachycardia, which were often triggered by taking a shower. These symptoms persisted for the next 6 years without an explanation or an identifiable etiology, despite receiving evaluations from a variety of specialists. Due to a severe fear of the sight of blood, anxiety, and obsessive–compulsive disorder, which was diagnosed by a primary care physician, she had to stop working and became housebound after the second pregnancy. She denied depressive symptoms or suicidal ideations. She did report Raynaud’s phenomenon as well as recurrent lower-back pain for at least 15 years, which was triggered by prolonged sitting. She was diagnosed with hypermobility spectrum disorder, though she denied chronic muscle or joint pain. She did experience easy bruising and bruxism. Prior to her second pregnancy, she used to run for at least 30 min several times per week. After her second pregnancy, she developed significant exercise intolerance due to resting and exertional tachycardia. She described having a heart rate of 150 beats per minute after walking for only 5 min. Sertraline 200 mg was initiated for OCD and anxiety, which resulted in a partial improvement in psychiatric symptoms; as such she was well enough to briefly leave the house. She reported experiencing chronic insomnia for many years, sleeping only 3 to 5 h per night. Clonazepam was prescribed at a dose of 0.5 mg to 1 mg at bedtime, but it did not prolong her sleep duration. She experienced frequent flushing and angioedema, sometimes triggered by stress, but most of the time, the trigger was unknown. She could not identify any potential food triggers. A gluten-free, low-histamine diet failed to help. After receiving an intramuscular cortisone injection for the treatment of a rash, she felt significantly better for several weeks, both regarding physical and psychological symptom improvement. At the time of her presentation, her most disabling symptoms were obsessive thoughts regarding her fear of seeing blood, fatigue, exercise intolerance, resting and postural tachycardia, and recurrent pruritic facial and neck rash.

A tilt table test demonstrated inappropriate sinus tachycardia (IST) and neurocardiogenic syncope after a nitroglycerin challenge. Diagnostic tests were also remarkable, showing low serum ferritin level, elevated platelets, and mildly reduced IgG1 subclass. Serum and urine MC mediators, including serum tryptase, serum histamine, serum and urine chromogranin, and prostaglandins, were in the normal range at baseline testing, during which she was not experiencing a flare.

Given the clinical features of allergic symptoms and excellent response to antihistamines and steroids, a clinical diagnosis of MCAS was made by an allergist based on the Consensus 2 criteria [6]. Daily hydroxyzine at a dose of 25 mg was initiated and increased to twice a day, along with 10 mg cetirizine daily. The patient reported significant improvement and near complete resolution of both her phobia of blood and obsessive–compulsive thoughts and rituals. Additionally, her elevated resting and exertional heart rate decreased after the implementation of antihistamines without the use of heart-rate-controlling medications typically used for the treatment of IST and other autonomic disorders, such as beta blockers or an I-channel blocker. Given her physical and psychological improvement, she was able to return to work in healthcare full time and resume an exercise training program.

### 3.2. Illustrative Case: Patient 4—Personal Account

“Here is my story of a lifelong battle with depression, mood swings and healing after a diagnosis and treatment of MCAS. I had depressive symptoms as a little girl, which escalated after I was molested at the age of 11. By the time I was 16, I tried to kill myself. In my early twenties I escalated to severe mood swings going into mania for 6 to 7 days at a time never sleeping or even lying down to an inability to stay awake for days on end. I would go into a rage at the drop of a hat and at times would lock myself in the bathroom to keep away from my sons so they would not receive the outcome of my rage. There was even a time I thought I should give my boys up for adoption because I believed they deserved a better mother than I was able to be. Along with the mood swings I also developed severe panic attacks”.

“I saw a psychiatrist who diagnosed me with bipolar disorder. I then started taking medications which would help a little for a short time, but then I’d be right back where I started. I was eventually put on Lithium that helped the symptoms, but I developed severe swelling in my abdomen to the point I had to wear maternity clothes and vomited around the clock. My mood and desire to die was also related to severe pain from a torn lumbar disc that went undiagnosed for 22 years. Between the pain and depression, I continued to get worse, until I attempted suicide again in 1990. I continued to try different medications but had drug reactions and a never-ending circle with depression and thoughts of suicide wrapped in the middle”.

“At age 55 I was diagnosed with breast cancer and underwent a mastectomy and chemotherapy. Shortly after chemo I started having multiple symptoms, including heart palpitations up to 200 bpm just walking across the room as well as chest pain and dangerously low blood pressure, I was finally diagnosed with POTS. I also had esophageal spasms, interstitial cystitis, migraine headache and episodes of severe vomiting. I also started suffering severe shortness of breath and was diagnosed with asthma and vocal cord dysfunction. I kept asking for help from doctors but was told it was all in my head and I just needed to see a psychiatrist! I knew all of this was not in my head, but when you have a history of depression and bipolar disorder, I found that very few doctors take you seriously”.

“My primary care physician finally referred me to a neurologist for POTS, and my life changed at that appointment when he diagnosed me with MCAS. He assured me my symptoms were not in my head but in my brain. He started me on the MCAS protocol, and although some symptoms were better the depression was not. He then referred me to Dr. Weinstock who tried other medications and eventually added a chemotherapy drug called hydroxyurea. Shortly after starting hydroxyurea, my suicidal thoughts finally stopped, and I have not had any desire to die since. I am truly happy for the first time in my life”.

## 4. Discussion

We present a case series of eight patients with refractory neuropsychiatric disorders who experienced significant improvement after subsequent diagnosis and treatment of MCAS. Previously, these patients exhibited either no response or poor response to a variety of psychiatric medications and psychotherapies. Electroshock therapy had been used in the eldest patient in the cohort. These patients often presented to psychiatrists in their early teenage years, but some developed neuropsychiatric symptoms as an adult, with pregnancy or the postpartum period being a precipitating or exacerbating event. The female predominance in our case series was similar to other MCAS studies where the female to male ratio was over 80% [23,24]. Most of the patients in our case series responded to simple Step 1 MC-directed therapy. The eldest of our patients was a 71-year-old woman who had experienced severe, lifelong depression. Low-dose hydroxyurea (500 mg per day) was added to her regimen to treat refractory gastrointestinal symptoms. To her surprise, her daily suicide ideation resolved for the first time in four decades. Hydroxyurea has been used successfully to treat general systemic symptoms in both refractory MCAS and mastocytosis [22]. This drug is an oral ribonucleotide reductase inhibitor which is used at a high dosage in the treatment of chronic myeloproliferative neoplasms and at a low dosage in sickle cell anemia, where there is evidence that MC activation causes increased cytokines and joint pain [25].

## 5. Mast Cells

We theorize that MCAS-associated neuropsychiatric disorders could be caused by abnormal MCs in the central and/or peripheral nervous system or indirectly by circulating MC mediators that lead to inflammation in the nervous system. MCs, known as immune and pro-inflammatory effector cells, are present in the meninges and are implicated in the pathophysiology of migraine via neuropeptide release, vasodilation, and plasma and protein extravasation, which can lead to MC degranulation. Since MCs release hundreds of various mediators, including histamines, tryptases, and leukotrienes, the degranulation of meningeal MCs contributes to the sensitization of trigeminal vascular afferent processing. This MC-mediated pathway is thought to be one of the mechanisms underlying migraine pain pathophysiology, and migraine is one of the most common comorbidities noted in patients with MCAS [26].

Similar to our case series, patients with MCAS can have comorbid autonomic dysfunction. While the mechanisms have not been explored in detail, a recent study linked the parasympathetic nervous system and MCs via its findings, which suggest that the endogenous acetylcholine activates the meningeal MCs [27]. Further studies are needed to delineate the complex interplay between the autonomic nervous system, MCs, and the connective tissues of the meninges, cerebral vasculature, and other structures important to the pathophysiology of the triad of dysautonomia, MCAS, and hypermobility spectrum disorders often observed in clinical practice [28].

Additionally, circulating autoantibodies could affect the brain and autonomic nervous system due to an MC-induced hyperpermeable BBB and/or an abnormally functioning blood–cerebrospinal fluid barrier [8,29]. The role of MC activation in a variety of neuropsychiatric disorders has been studied in humans and in animal models [30,31,32,33,34,35,36,37]. Magnetic resonance imaging has demonstrated morphological and functional abnormalities in the brains of mastocytosis patients with neuropsychiatric complaints [38]. Using the same technique, a MCAS patient with depression also exhibited the same radiographic finding [39]. In a case series of 139 patients with mastocytosis, 49% had depression [31]. In another series of 288 mastocytosis patients, the prevalence of depression was 64%, with 56% described as having moderate and 8% severe depression [30].

## 6. Histamine and Histamine Receptors

Histamine, a major MC mediator, is a known neurotransmitter in the central nervous system. Histamine is a monoamine that is metabolized from the precursor histidine and is released into some of the neuronal synapses, as well as into the blood stream, where it acts as a hormone. Histamine is also a known neuromodulator since it regulates the release of other neurotransmitters, such as acetylcholine, norepinephrine, and serotonin [40]. The histamine receptors H1, H2, H3, and H4 are a class of G-protein–coupled receptors which bind to histamine as their primary endogenous ligand [41,42]. The H1 receptor mediates immediate hypersensitive reactions, such as wheezing, itching, coughing, and hypotension; the H2 receptor affects gastric mucosa, vascular smooth muscle, fat cells, basophils, and neutrophils and inhibits antibody synthesis, T-cell proliferation, and cytokine production; the H3 receptor decreases the release of acetylcholine, serotonin, and norepinephrine neurotransmitters in the central nervous system; and the H4 receptor is implicated in mast cell chemotaxis and regulating immune responses [42,43,44].

Histamine is known to contribute to the regulation of sleep and wakefulness, and histamine blocking is a well-known pharmacological approach used to induce sleep. Low levels of histamine have been shown to correlate with schizophrenia, and an altered histaminergic system has been found in the nigrostriatal network in Parkinson’s disease [40]. Postmortem studies have revealed alterations in the histaminergic system in neurological and psychiatric diseases. Brain histamine levels are decreased in Alzheimer’s disease patients, whereas abnormally high histamine concentrations are found in the brains of Parkinson’s disease and schizophrenic patients [40]. Low histamine levels are associated with convulsions and seizures [40]. The release of histamine is altered in response to different types of brain injury; for example, the increased release of histamine in an ischemic brain trauma might play a role in recovery following neuronal damage [43]. Neuronal histamine is also involved in pain, and drugs that increase brain and spinal histamine concentrations have antinociceptive properties. Histaminergic drugs, most importantly histamine H3 receptor ligands, have shown efficacy in many animal models of the neurologic disorders, and clinical trials to determine the efficacy and safety of these drugs in humans are needed [43].

Histamine has a significant underexplored potential to provide targets for many CNS disorders. The histamine system has been suggested as a possible target for the treatment of psychiatric disorders, and drugs that modulate this system have been proposed as cognitive enhancers [44]. Greater understanding of histamine, histamine receptors, and histaminergic pathways in the central and peripheral nervous systems is particularly relevant for the development of novel pharmacological treatments for neurologic and psychiatric disorders [42,45,46,47].

Lastly, histamine may cause the increased permeability of the blood–brain barrier. It also significantly influences neuroendocrine control, including the behavioral state, biological rhythms, energy metabolism, thermoregulation, fluid balance, stress, and reproduction. In addition to being a neurotransmitter and neuromodulator, histamine is also associated with the functioning of the immune system. During an immune reaction, histamine is released and contributes to the physiologic changes necessary for the immune system to fight a pathogen, including an increase in blood pressure, temperature, swelling, and bronchial constriction [8,42,45,47].

## 7. Mast-Cell-Directed Treatment

Standard psychiatric medicines are frequently prescribed for patients presenting with depression and anxiety; however, a significant subset of patients is refractory to these treatments or experiences adverse events. Our case series suggests that when MCAS is suspected and then diagnosed, MC-directed therapy can be effective in improving neuropsychiatric manifestations. Treatment with antihistamines, MC-stabilizing agents, and other pharmacologic modalities such as LDN, along with non-pharmacologic approaches including avoiding symptomatic triggers and adopting low-histamine and gluten/dairy-free diets, can be effective, are inexpensive, and have a low side effect profile compared to standard antidepressant and antianxiety therapies. Another consideration regarding the intolerance to standard psychiatric medications experienced by the general population, particularly for MCAS patients, is the frequency of reaction to excipients [48].

Benzodiazepines are used in the treatment of anxiety and panic attacks. These medications have been demonstrated to have an inhibitory action on the pro-inflammatory effector functions of MCs [49]. In addition, there is evidence in the literature that supports the role of histamine as a neurotransmitter in stress-related disorders [50]. Microglia express all four histamine receptors, with selective upregulation of H1R and H4R [51]. Astrocytes express the H1R and H2R histamine receptors [52]. This may be another mechanism for the effect of histamine receptor blockers in MCAS and psychiatric disorders. Nevertheless, due to the adverse effects and addictive potential associated with chronic benzodiazepine use, benzodiazepines should not be routinely prescribed and should be reserved for patients who are refractory to other non-addictive MC therapies.

For severe cases of MCAS, immune modulators can be helpful. These medications include hydroxyurea and various tyrosine kinase inhibitors (TKI). Masitinib, a TKI, has been used as effective treatment for drug-refractory depression in mastocytosis and is currently being studied in MCAS patients [30,53] [Clintrials.gov NCT05449444]. In the largest mastocytosis case series to date, of 288 patients treated with masitinib, 67% experienced a significant improvement with regard to depression overall, and 75% recovered [30]. In a case report, a MCAS patient with severe MCAS and postural orthostatic tachycardia syndrome experienced significant improvement with regard to depression, anxiety, and dysautonomia symptoms using intravenous immunoglobulin, LDN, and antibiotic treatment for small intestinal bacterial overgrowth [21].

## 8. Autonomic Dysfunction and MCAS

Autonomic dysfunction is common in MCAS patients, possibly due to the MC mediator effects on the central autonomic networks in the brain, peripheral autonomic and small nerve fibers, and the vasodilatory effects of the mediators on blood vessels and via other yet unidentified mechanisms. One study identified clinical evidence of MC hyperactivity in 64% of their patients with POTS, 66% of whom received at least one positive laboratory finding suggestive of MC hyperactivity [28]. In another study, the percentage of MCAS diagnosis within a group of POTS and hEDS patients was 31% in comparison with 2% in a group without POTS or hEDS [54]. Small fiber neuropathy and cerebral hypoperfusion may share pathophysiologic mechanisms in MCAS, dysautonomia, and hEDS [55]. While the true prevalence of MCAS in patients with POTS or hEDS is unknown, considering the lack of awareness of MCAS as a diagnostic entity among clinicians and the difficulty of confirming diagnosis objectively, most of the patients in our series had comorbid autonomic disorders, with POTS being the most common diagnosis.

## 9. Limitations

The limitations of this study include those inherent to the nature of a retrospective chart review, the small sample size, the subjectivity of patient-reported functional improvement following treatment, the lack of a control group, the referral bias and heterogeneity of various mast-cell-directed treatment, the influence of psychiatric medications, and the treatment approaches for autonomic comorbidities and other comorbidities. In addition, we recognize that there is some controversy regarding the diagnostic criteria for MCAS [6]. There are limitations related to our patient selection and the generalizability of our findings.

## 10. Clinical Relevance in Personalized Care

Although the literature on MCAS, dysautonomia, and hypermobility spectrum disorders is relatively limited, patients who have all three conditions as a triad are often encountered in clinical practice [56]. Many of these patients have been sick for years or decades, seen multiple physicians of various specialties, tried a wide variety of medications and supportive therapies with limited improvement, and have experienced significant functional impairment. While neurologic, autonomic, and psychiatric comorbidities in these patients are numerous, many patients are misdiagnosed with psychiatric diagnoses such as somatic symptom disorder, medically unexplained symptoms, functional neurologic disorders, somatization disorder, factitious disorder, or malingering. Some parents of children and teens with the triad have been wrongfully accused of Munchausen by proxy. The mislabeling of these patients with psychiatric illness as the cause for a systemic illness often leads to inappropriate or misdirected treatment, iatrogenic adverse events, resentment, mistrust on the part of the patient, doctor shopping, non-compliance, medical care avoidance, and psychological symptoms and trauma caused by their negative experience with the healthcare system. As the illustrative cases demonstrate, significant improvement in and even resolution of decades of neuropsychiatric symptoms are possible when an underlying systemic disorder is identified and therapeutic modalities for the underlying systemic disorder are instituted. Although at this time it is unknown whether a relationship between MCAS, dysautonomia, and hypermobility spectrum disorders is rooted in causation or association, we believe that a patient-centered, comprehensive, and personalized approach to neurologic and psychiatric care is essential to accurate diagnosis, effective treatment, and reducing the symptom burden and disability associated with these multisystemic complex chronic disorders.

## 11. Conclusions

In patients with neuropsychiatric disorders refractory to standard therapy who have systemic symptoms, underlying MCAS should be considered in the differential diagnosis. This is especially the case if they have comorbid POTS, other types of dysautonomia, and/or hEDS. Prospective randomized controlled trials are needed to determine the prevalence of MCAS in patients with treatment-refractory neuropsychiatric disorders, delineate the neurologic and psychiatric manifestations of MCAS, and assess the response to MCAS-targeted treatment.

## Figures and Tables

**Figure 1 jpm-13-01562-f001:**
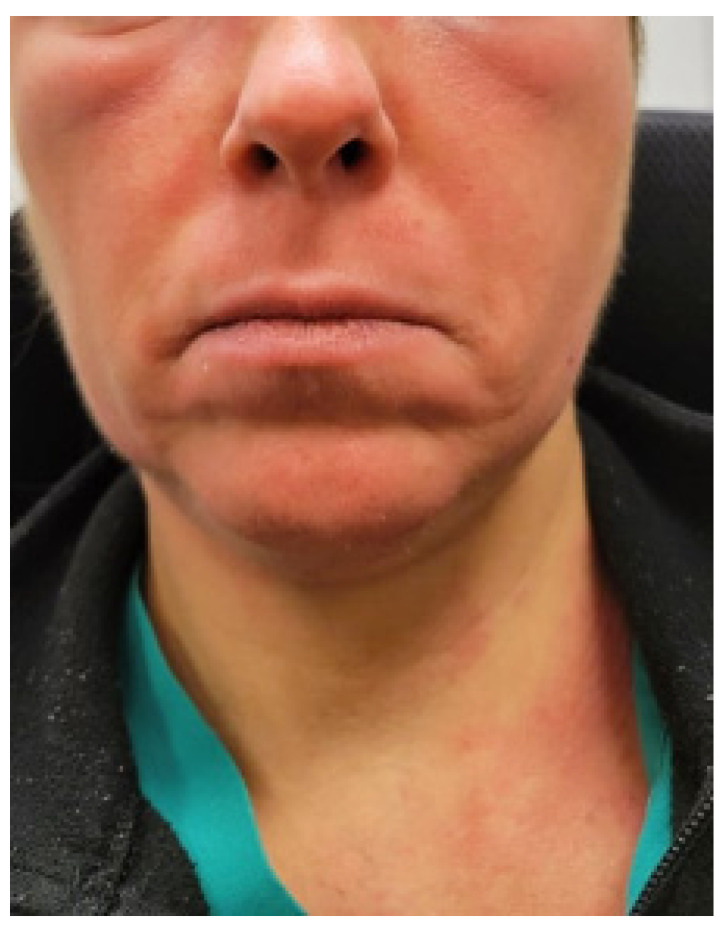
Photo of the periorbital edema with facial and neck flushing provided by patient 1.

**Table 1 jpm-13-01562-t001:** Common symptoms of mast cell activation syndrome. There are many heterogenous phenotypes that vary according to mast cell location, number, and ability to degranulate specific mediators. Symptoms may be continuous or intermittent and be of various levels of severity.

Constitutional	Fatigue, subjective hyperthermia and/or hypothermia, sweats, change in appetite, weight gain/loss, chemical/physical sensitivities, poor healing
Dermatologic	Urticaria, itch, flushing, hemangiomas with itch/pain, various rashes, telangiectasias, striae, skin tags, folliculitis, ulcers, eczema, angioedema, alopecia, onychodystrophy
Ophthalmologic	Irritated, “dry” eyes, difficulty focusing, blepharospasm
Otologic	Tinnitus, hearing loss, coryza, rhinitis, nasal congestion, epistaxis
Oropharyngeal	Pain, burning, leukoplakia, ulcers, angioedema, dysgeusia, dental and/or periodontal inflammation/decay
Lymphatic	Lymphadenopathy, rare splenomegaly
Pulmonary	Dry cough, dyspnea (difficulty taking a deep breath), wheezing, obstructive sleep apnea
Cardiovascular	Presyncope, hypertension, blood pressure lability, palpitations, edema, chest pain, allergic angina (Kounis syndrome)
Gastrointestinal	Dyspepsia, gastroesophageal reflux, abdominal pain, nausea, vomiting, diarrhea and/or constipation, gastroparesis, angioedema, dysphagia (usually proximal), bloating (post-prandial or spontaneous), malabsorption
Genitourinary	Menorrhagia, pelvic pain, endometriosis, vulvodynia, vaginitis, dysmenorrhea, miscarriages, infertility, dysuria
Musculoskeletal	Myalgias, migratory bone/joint pain, osteopenia/osteoporosis
Neurologic	Headache, migraine, sensory neuropathies, dysautonomia, episodic weakness, seizure disorders, non-epileptic seizures, cognitive dysfunction, insomnia, hypersomnolence, restless leg syndrome
Psychiatric	Depression, anger/irritability, mood lability, anxiety, panic, obsession–compulsion, attention deficit/hyperactivity
Hematologic	Easy bruising, polycythemia, anemia
Immunologic	Hypersensitivity reactions, increased risk for malignancy and autoimmunity, impaired healing, increased susceptibility to infection

**Table 2 jpm-13-01562-t002:** Diagnostic criteria for mast cell activation syndrome.

Consensus 1:
Severe, recurrent mast cell symptoms, which often include anaphylaxis and involve 2 or more organs, including urticaria, flushing, pruritus, angioedema, nasal congestion/pruritus, wheezing, throat swelling, hoarseness, headache, hypotensive syncope, tachycardia, cramping, and diarrhea; Increased mast cell mediators “preferably tryptase or increased tryptase from baseline plus 20% + 2 during an attack” or “less specific mediators” (plasma histamine or prostaglandin D2, serum heparin, urine N-methylhistamine); Response to mast-cell-directed therapy.
Consensus 2:
Presence of 2 systems with typical mast cell activation symptoms and ≥1 of the following: Positive mast cell mediators (plasma histamine or prostaglandin D2, serum heparin, tryptase, and chromogranin A, or urine N-methylhistamine, leukotriene E4, and 2, 3 dinor prostaglandin F2 alpha); Biopsy showing >20 mast cells per high-power field; Positive clinical response to mast-cell-directed therapy

**Table 3 jpm-13-01562-t003:** Neuropsychiatric manifestations of mast cell activation syndrome patients and response to mast-cell-directed therapy: cases 1–4. Neuropsychiatric manifestations of mast cell activation syndrome and response to mast-cell-directed treatment: cases 5–8.

**N**	**1**	**2**	**3**	**4**
Age (years), sex	47, female	50, female	37, female	71, female
Prior psychiatric diagnoses	GAD, OCD, phobia	GAD, panic disorder	Bipolar disorder (suicide attempt age 15), GAD, ADHD, Tourette’s, narcolepsy	MDD (suicide attempt age 16)
Clinical course in childhood and adolescence	Anaphylaxis to nuts and antibiotics	None	Brain fog, diarrhea, urticaria, self-abusive behavior, asthma	Headaches, recurrent viral infections, hives, edema with insect bites, allergies, nausea, abdominal pain, depression, menorrhagia
Clinical course in adulthood	Postpartum phobias, rashes, facial swelling, pruritus, syncope, tachycardia, migraine	Syncope/presyncope during pregnancy, pacemaker for bradycardia, tachycardia, blurred vision, anxiety, joint pain	Depression (daily suicidal ideation), mania, hallucinations, anxiety, fatigue, abdominal pain, nausea, myalgia, hives, bone pain, episodic hypertension, bedridden 4 days/week	Depression (daily suicidal ideation), pelvic pain leading to hysterectomy age 21, tinnitus, chest and body pain, interstitial cystitis
Prior psychiatric therapy	Multiple SSRIs without efficacy	Prescribed SSRI: elected not to take it	1 SSRI, 2 SSRNIs, 2 anti-psychotics, 3 benzos, lamotrigine, atomoxetine, dextroamphetamine, guanfacine	3 classes of anti-depressants—multiple agents, lithium, and ECT
New diagnoses	MCAS, hEDS, NCS, IST	MCAS, NCS, labile hypertension	MCAS, POTS, RLS, labile hypertension	MCAS, POTS
Mast cell treatment	Hydroxyzine, cetirizine daily. Prednisone PRN flares	Cetirizine and famotidine daily.	Step 1 therapy, LDN. Maintained on aripiprazole, dextro-amphetamine, and lamotrigine	Antihistamines 1 and 2, hydroxyurea
Outcomes of mast cell treatment on neuropsychiatric conditions	Complete response: works full time	Complete response: works full time. Tachycardia, syncope, flushing, and anxiety resolved	Partial response: works part time	Complete response: independent in ADLs and iADLs. Depression resolved
**N**	**5**	**6**	**7**	**8**
Age (years), sex	18, male	18, female	19, female	33, female
Prior neuropsychiatric diagnoses	Panic disorder, GAD, MDD	Panic disorder, GAD, MDD	Panic disorder, MDD	Panic disorder, GAD, MDD
Clinical course in childhood and adolescence	Brain fog, fatigue, rhinitis, diarrhea, abdominal pain with gluten	Constipation, diarrhea, dysphagia, heartburn, nausea, eczema, headache, menorrhagia, syncope	Nausea, diarrhea, menorrhagia, flushing, fatigue, brain fog, tinnitus	Headache, multiple viral infections
Clinical course in adulthood	Myalgias	Constipation, diarrhea, dysphagia, heartburn, nausea, eczema, headache, menorrhagia, syncope	Weight loss, nausea, diarrhea, menorrhagia, flushing, fatigue, brain fog, tinnitus	Nausea, pain, fatigue, weakness, tinnitus, palpitations, flushing, presyncope, migraine, brain fog, hives, itch, bone pain
Prior psychiatric therapy	Escitalopram	Escitalopram, buspirone	Desvenlafaxine, fluvoxamine, fluoxetine	None
New diagnoses	MCAS	MCAS, RLS, hEDS	MCAS, POTS, hEDS	MCAS, POTS, hEDS
Mast cell treatment	Step 1, LDN	H1/2 blockers, LDN, buspirone PRN anxiety	Step 1, LDN	GFD, Step 1, LDN
Outcome on mast cell treatment for neuropsychiatric conditions	Complete response: Able to return to college after withdrawal	Complete response: Able to attend college after home schooling	Marked improvement: Able to return to college. Regained 15 pounds	Complete response: Able to work full time

Abbreviations: ADHD, attention-deficit/hyperactivity disorder; benzos, benzodiazepines; ECT, electroconvulsive therapy; GDF, gluten-free diet; GAD, generalized anxiety disorder; hEDS, hypermobile Ehlers-Danlos syndrome; IST, inappropriate sinus tachycardia; LDN, low-dose naltrexone; MCAS, mast cell activation syndrome; MDD, major depressive disorder; NCS, neurocardiogenic syncope; OCD, obsessive–compulsive disorder; POTS, postural orthostatic tachycardia syndrome; PRN, as needed; SSRI, selective serotonin reuptake inhibitor. Note: Step 1 therapy is a combination mast-cell-directed oral therapy using various over-the-counter histamine 1 receptor antagonists (variety of brands, twice daily), histamine 2 receptor antagonist (famotidine 20 mg twice daily), quercetin (1000 mg twice a day), sustained release vitamin C (1000 mg daily), and vitamin D (2000–5000 mg daily depending on vitamin D level).

## Data Availability

All of the patient’s histories are maintained in the electronic medical records of Blitshteyn and Weinstock.

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
