# Peer review of "Neuropsychiatric Manifestations of Mast Cell Activation Syndrome and Response to Mast-Cell-Directed Treatment: A Case Series"

_jpm, 2023, doi:10.3390/jpm13111562_

Round 1

Reviewer 1 Report

Comments and Suggestions for Authors

The manuscript entitled "Neuropsychiatric manifestations of mast cell activation syndrome and response to mast cell directed treatment: a case series" presents case series of a rare condition, which makes the topic very interesting. 

I have several recommendations to the authors:

1. It would be better if in the title include " and systematic review of the literature"

2. Include in the introduction part some information about mast cells and histamine ( and eliminate it from the discussion).

3. Remove some of the key words - they are too many ( they could be summarized in  "neuropsychiatric manifestations")

4. Clarify what kind of medications are taken by the patients?  -(Administration of additional medicines including immune modulators are used for refractory MCAS)

5. Describe briefly the other cases.  It is appropriate to be presented, because the manuscript is entitled - case series. 

6. My recommendation to the authors is to use some figures to present the histamine, histamine receptors and also clinical presentations.

7. In addition, If some pictures of the rash of the patients are available, they can include them in the manuscript.  

8. Make the conclusion shorter and clear. 

9. Check the references and include DOI, where it is possible.

Finally, I can add, that the manuscript is well-written, very interesting and a good idea to be presented like a case series.  

Comments on the Quality of English Language

The English language is on sufficient level. Minor revision.

Author Response

Response to Reviewer #1

Reviewer 1:

The manuscript entitled "Neuropsychiatric manifestations of mast cell activation syndrome and response to mast cell directed treatment: a case series" presents case series of a rare condition, which makes the topic very interesting.

I have several recommendations to the authors:

  1. It would be better if in the title include " and systematic review of the literature"

      We are unable to change the title to include the phrase “systematic review of literature” because we did not conduct review of literature as this was a case series. The literature presented in the paper was not selected systematically.

  1. Include in the introduction part some information about mast cells and histamine (and eliminate it from the discussion).

     A paragraph was moved up on MCs from the Discussion into the Introduction. We left the information about histamine in the Discussion since it had data and theoretical information. Histamine is only one of 1000 mediators that mast cells can produce.

  1. Remove some of the key words - they are too many ( they could be summarized in "neuropsychiatric manifestations")

   Original  Keywords: Anxiety; bipolar; depression; dysautonomia; postural orthostatic tachycardia syndrome; POTS; MCAS; mast cell activation syndrome; panic disorder

  Altered   Keywords: Anxiety; depression; ; neuropsychiatric, dysautonomia, POTS,; panic disorder

  1. Clarify what kind of medications are taken by the patients? -(Administration of additional medicines including immune modulators are used for refractory MCAS)

      Please see details in Tables 3 a and b. We thought it would be best to summarize accordingly.

  1. Describe briefly the other cases. It is appropriate to be presented, because the manuscript is entitled - case series.

       We were concerned about length of the paper and thought the reader would enjoy 2 in depth and the rest in summary fashion as per the table.

  1. My recommendation to the authors is to use some figures to present the histamine, histamine receptors and also clinical presentations.

We have added Figure 1

 Since the paper is not a review, we decided to concentrate on the patient presentations and clinical care. We think a figure with histamine and receptors would be more appropriate for a review paper, which this is not.

  1. In addition, If some pictures of the rash of the patients are available, they can include them in the manuscript.

      A photo of flushing and periorbital swelling for patient 1 was added as Figure 1.

  1. Make the conclusion shorter and clear.

We shortened the conclusion significantly to make it more concise.

  1. Check the references and include DOI, where it is possible.

Done – will reorder the references on the clean version.

Finally, I can add, that the manuscript is well-written, very interesting and a good idea to be presented like a case series.
   Thank you for your comments.

Reviewer 2 Report

Comments and Suggestions for Authors

In this study, the Authors present a retrospective case series of 8 patients (7 females, 1 male, with a mean age of 36 years, range 18 to 71 years) diagnosed with refractory neuropsychiatric disorders who received subsequently the diagnosis of Mast cell activation syndrome (MCAS) and improved with MC-directed treatment. Prior neuropsychiatric diagnoses that were diagnosed, medical therapy, past medical history, and outcomes with MC-directed therapy are presented and discussed.

The study is of clinical interest and original findings. However, to further improve the clinical impact, some points should be addressed.

-According to literature data, in a reasonable number of cases, signs and symptoms of MCA can be detected, but the criteria of MCAS could not be fulfilled. There are, in fact, patients in whom the symptoms are less severe and/or restricted to one organ system or even a local organ site. They may be suffering from food-intolerance, drug side effects, toxin exposure, or autoimmune disease as previously reported (Mast cell activation syndrome: Importance of consensus criteria and call for research. J Allergy Clin Immunol. 2018 Sep;142(3):1008-1010.). In particular, the authors should recall literature data demonstrating the significant associations of various neurological (and neuropsychiatric) disorders in some autoimmune diseases such as celiac disease that may be characterized by the positivity of circulating autoantibodies to neuronal antigens as previously demonstrated (Sera of patients with celiac disease and neurologic disorders evoke a mitochondrial-dependent apoptosis in vitro. Gastroenterology. 2007 Jul;133(1):195-206.) or autoantibodies to gangliosides, as previously demonstrated (Anti-ganglioside antibodies and celiac disease. Allergy Asthma Clin Immunol. 2021 May 28;17(1):53. doi: 10.1186/s13223-021-00557-y. ).

-Regarding the autoimmune disease, and circulating autoantibodies, the authors should further describe the reason for the gluten-free diet of patient N. 8 (as reported in Table 3). Why she was on a gluten-free diet? If a diagnosis of celiac disease was previously done, it should be reported as well as the diagnostic criteria satisfying the current international guidelines (Current guidelines for the management of celiac disease: A systematic review with comparative analysis. World J Gastroenterol. 2022 Jan 7;28(1):154-175.).

-The authors properly address the potential role of circulating autoantibodies that could affect the brain and autonomic nervous system due to a MC-induced hyperpermeable BBB. It has been previously suggested a pathogenic role of circulating autoantibodies for example in celiac disease patients with neurological disorders and seropositivity for antinueronal autoantibodies able to evoke a mitochondrial-dependent apoptosis (Sera of patients with celiac disease and neurologic disorders evoke mitochondrial-dependent apoptosis in vitro. Gastroenterology. 2007;133:195-206.) and other autoantibodies as a consequence of the "secondary autoimmunity" hypothesis as previously reported (Anti-actin IgA antibodies in severe coeliac disease. Clin Exp Immunol. 2004 Aug;137(2):386-92. doi: 10.1111/j.1365-2249.2004.02541.x.

Author Response

Response to Reviewer 2:

In this study, the Authors present a retrospective case series of 8 patients (7 females, 1 male, with a mean age of 36 years, range 18 to 71 years) diagnosed with refractory neuropsychiatric disorders who received subsequently the diagnosis of Mast cell activation syndrome (MCAS) and improved with MC-directed treatment. Prior neuropsychiatric diagnoses that were diagnosed, medical therapy, past medical history, and outcomes with MC-directed therapy are presented and discussed.

The study is of clinical interest and original findings. However, to further improve the clinical impact, some points should be addressed.

  1. According to literature data, in a reasonable number of cases, signs and symptoms of MCA can be detected, but the criteria of MCAS could not be fulfilled. There are, in fact, patients in whom the symptoms are less severe and/or restricted to one organ system or even a local organ site. They may be suffering from food-intolerance, drug side effects, toxin exposure, or autoimmune disease as previously reported (Mast cell activation syndrome: Importance of consensus criteria and call for research. J Allergy Clin Immunol. 2018 Sep;142(3):1008-1010.). In particular, the authors should recall literature data demonstrating the significant associations of various neurological (and neuropsychiatric) disorders in some autoimmune diseases such as celiac disease that may be characterized by the positivity of circulating autoantibodies to neuronal antigens as previously demonstrated (Sera of patients with celiac disease and neurologic disorders evoke a mitochondrial-dependent apoptosis in vitro. Gastroenterology. 2007 Jul;133(1):195-206.) or autoantibodies to gangliosides, as previously demonstrated (Anti-ganglioside antibodies and celiac disease. Allergy Asthma Clin Immunol. 2021 May 28;17(1):53. doi: 10.1186/s13223-021-00557-y. ).

Answer: Thank you for your thoughtful comments. None of the patients in our case series had celiac disease.

  1. Regarding the autoimmune disease, and circulating autoantibodies, the authors should further describe the reason for the gluten-free diet of patient N. 8 (as reported in Table 3). Why she was on a gluten-free diet? If a diagnosis of celiac disease was previously done, it should be reported as well as the diagnostic criteria satisfying the current international guidelines (Current guidelines for the management of celiac disease: A systematic review with comparative analysis. World J Gastroenterol. 2022 Jan 7;28(1):154-175.).

Answer:   

From the Materials and Methods: “As part of our standard approach, we advise a 3-week trial on a gluten-free, dairy-free, and low histamine diet.” Many of our patients are sensitive to one or more foods. Gluten particularly is a common trigger - most often without celiac disease.

3 .  The authors properly address the potential role of circulating autoantibodies that could affect the brain and autonomic nervous system due to a MC-induced hyperpermeable BBB. It has been previously suggested a pathogenic role of circulating autoantibodies for example in celiac disease patients with neurological disorders and seropositivity for antinueronal autoantibodies able to evoke a mitochondrial-dependent apoptosis (Sera of patients with celiac disease and neurologic disorders evoke mitochondrial-dependent apoptosis in vitro. Gastroenterology. 2007;133:195-206.) and other autoantibodies as a consequence of the "secondary autoimmunity" hypothesis as previously reported (Anti-actin IgA antibodies in severe coeliac disease. Clin Exp Immunol. 2004 Aug;137(2):386-92. doi: 10.1111/j.1365-2249.2004.02541.x.)

Answer:

Thank you for your perspective on celiac disease. We agree that it’s possible that patients with MCAS have circulating antibodies targeting neuronal antibodies; however, research on antibodies in MCAS is extremely limited, which is why we conclude the paper by calling for more research studies on this very important topic.